# Zgli: A Pipeline for Clustering by Compression with Application to Patient Stratification in Spondyloarthritis

**DOI:** 10.3390/s23031219

**Published:** 2023-01-20

**Authors:** Diogo Azevedo, Ana Maria Rodrigues, Helena Canhão, Alexandra M. Carvalho, André Souto

**Affiliations:** 1LASIGE, Departamento de Informática da Faculdade de Ciências, Universidade de Lisboa, 1749-016 Lisboa, Portugal; 2EpiDoC Unit, The Chronic Diseases Research Centre, NOVA Medical School, NOVA University of Lisbon, 1169-056 Lisboa, Portugal; 3Comprehensive Health Research Center, NOVA Medical School, NOVA University of Lisbon, 1150-082 Lisboa, Portugal; 4Instituto de Telecomunicações, 1049-001 Lisboa, Portugal; 5Department of Electrical and Computer Engineering, Instituto Superior Técnico, Universidade de Lisboa, 1049-001 Lisboa, Portugal; 6Lisbon Unit for Learning and Intelligent Systems, 1049-001 Lisboa, Portugal

**Keywords:** clustering by compression, normalized compression distance, Kolmogorov complexity, CompLearn, Zgli, clustering techniques

## Abstract

The normalized compression distance (NCD) is a similarity measure between a pair of finite objects based on compression. Clustering methods usually use distances (e.g., Euclidean distance, Manhattan distance) to measure the similarity between objects. The NCD is yet another distance with particular characteristics that can be used to build the starting distance matrix for methods such as hierarchical clustering or K-medoids. In this work, we propose Zgli, a novel Python module that enables the user to compute the NCD between files inside a given folder. Inspired by the CompLearn Linux command line tool, this module iterates on it by providing new text file compressors, a new compression-by-column option for tabular data, such as CSV files, and an encoder for small files made up of categorical data. Our results demonstrate that compression by column can yield better results than previous methods in the literature when clustering tabular data. Additionally, the categorical encoder shows that it can augment categorical data, allowing the use of the NCD for new data types. One of the advantages is that using this new feature does not require knowledge or context of the data. Furthermore, the fact that the new proposed module is written in Python, one of the most popular programming languages for machine learning, potentiates its use by developers to tackle problems with a new approach based on compression. This pipeline was tested in clinical data and proved a promising computational strategy by providing patient stratification via clusters aiding in precision medicine.

## 1. Introduction

Cluster analysis is a chief pattern recognition technique [1]. It is characterized by using resemblance or dissemblance measures between the objects to be identified. The pattern recognition aspect of clustering has been extensively used in healthcare over the years [2]. Clustering algorithms can find patterns across patients that are difficult for medical practitioners [3]. This approach has enabled many solutions for problems in diseases like Amyotrophic Lateral Sclerosis [4], Alzheimer’s disease, Parkinson’s Disease [5], rheumatic diseases [6], and cancer [7], to name a few.

Patient stratification by clustering them into groups allows us to identify which patients will benefit from what interventions before their condition worsens, being a leading step to precision medicine. Several stratification methods have successfully been applied in biomedical research and assisted the treatment process [8,9]. Because of the heterogeneity and complexity of medical data, it is very challenging to integrate them and use them in the practical clinic. There are two significant challenges: combining multiple data sources leads to complex problems, and the disparity of different data types causes a critical problem in machine learning for biomedical data.

Clustering can find patterns by grouping individuals based on similarity, and multiple types of distances can express this similarity. The most commonly used distances for these methods are geometric ones, such as the Euclidean distance, Manhattan distance, Minkowski distance, and Hamming distance [10]. This paper will focus on the normalized compression distance (NCD), a distance based on algorithmic information theory [11].

The NCD is a less explored way of computing the similarity between objects compared to other popular distances, such as Euclidean and Manhattan distances. Nevertheless, NCD has shown potential to be a sound approach in different fields of application, especially when dealing with a dataset comprised of multiple files. In music [11], NCD was able to cluster music data by different degrees of granularity, beginning in genre and narrowing it down to the author. Moreover, in computer science [12,13], the authors showed that cataloging network traffic and a computer virus is possible. It also has applications in computer security since, for example, one can break multi-party computations based on [14]. In biology, NCD was used to classify mitochondrial genome phylogeny [15]. In literature [16], NCD was able to recognize the similarity of languages. In medicine [17], NCD was used to cluster medical data, particularly fetal heart rate. In [18], the NCD was able to group different types of files very well, even when applied to data with high noise levels. Furthermore, in recent studies [19,20], the NCD was used to group similar RNA sequences of different viruses to find a plausible origin of SARS-CoV-2.

In this work, we propose Zgli [21], a new Python tool developed to enable users to cluster by compressing any given set of files and many different types of data. The choice of Python stands from it being one of the most popular programming languages in the computer science field [22]. Indeed, it is leading the pack in the machine learning community, being chosen by 57% of data scientists and machine learning developers using it and 33% prioritizing it for development [23].

The Zgli tool augments the current state-of-the-art tool for clustering by compression—the Complearn [11] tool. It iterates over some of the features already available in Complearn and simplifies the process of integrating a clustering by compression approach with the current most used tools for clustering and data science.

In this paper, we will present the new features implemented in the proposed Zgli tool that improves over Complearn, providing replicable test cases of these features in action. Additionally, we show a use case where we applied the Zgli framework to a real-world dataset comprised of clinical data and verified that it competes with standard clustering techniques. In particular, in the Zgli tool, we add a feature that is able to perform the NCD distance with compression by columns, and a new encoder for categorical data.

Finally, it is essential to note that all the documentation necessary to use Zgli is available in Section 4, accompanied by a quick start tutorial for the tool and shortcuts for the source code published on GitHub to any developer who wishes to edit it.

The rest of the document is organized as follows. In Section 2, we present the basic notions and background necessary to understand the topics discussed in this paper. Section 3 describes the Complearn tool and gives an overview of its features. Section 4 describes the Zgli in detail, particularly looking into its two main classes, Folder and Encoder. Section 5 presents the tests and the results achieved using the module’s new features. Finally, Section 7 outlines the conclusions attained in this work and discusses some possible topics for future work.

## 2. Background

In this section, we look at the background concerning two essential subjects to understand all the remaining sections in this paper. The first one is dedicated to the understanding of the basic concepts leading to NCD, namely Kolmogorov complexity and normalized information distance. The second is the quartet method employed by the Complearn tool to build clusters from a distance matrix. We refer the reader to the book of Li and Vitányi [24] for a complete and comprehensive introduction to the themes related to algorithmic information theory.

### 2.1. Kolmogorov Complexity

The Kolmogorov complexity (*K*) is a concept from algorithmic information theory and can be described as a measure of information in a string. The Kolmogorov complexity of an object is the length of the shortest program in a predetermined syntax (i.e., Turing Machine, Python, Java, Lisp) that, when run, produces that same object as output. So K(x) is the length of the most compressed program describing *x*:(1)$$K\left(x\right)=\min_{p}\left\{|p|:U\left(p\right)=x\right\},$$
where *U* is any universal (Turing) machine.

As an example that illustrates this concept, consider the following strings:
$s_{1}$: “ggggggggggggggggg”with size =17;$s_{2}$: “LATQvgkCQaNwEadqO”with size =17.

Although they have the same size, when compared in algorithmic terms, the information needed to describe the two strings, one can use the following programs $p_{1}$ and $p_{2}$ describing $s_{1}$ and $s_{2}$, respectively:
$p_{1}$: “return ’g’ * 17”with $K(s_{1})\approx$ 4;$p_{2}$: “return ’LATQvgkCQaNwEadqO’”with $K(s_{2})\approx$ 17.

Notice that the number of times ’g’ occurs in $s_{1}$ describes the information in $s_{1}$ is repeated. Therefore, the algorithmic information needed to describe $s_{1}$—described by $K(s_{1})$—is much shorter than writing the entire string. On the other hand, there are no “patterns” that one can use to express more succinctly the string $s_{2}$. Therefore, $K(s_{2})$ is nearly maximally, i.e., approximately equal to the size of $s_{2}$. It is well known that the Kolmogorov complexity is not computable, i.e., given a string *x*, determining K(x) cannot be determined by any automated machine. The alternative is to use approximations based on the size of the outcome of regular computational compressors like Winzip.

The following subsection explains how to use the (computable version of) Kolmogorov complexity to compare two strings.

### 2.2. Normalized Compression Distance

The normalized compression distance (NCD) measures the similarity between two objects, but to understand it, we first need to look at the concept of information distance and its normalized version. The information distance (ID) represents the distance between two objects *x* and *y* by the shortest program, which transforms *x* into *y* and vice-versa, given by:(2)ID(x,y)=max{K(x|y),K(y|x)}.

The information distance gives us the absolute distance between two objects, independently of the size of the objects. To avoid being influenced by the size of the objects, to measure the similarity between objects, we must first normalize this distance to obtain the relative distances between objects. For example, the ID can tell us that two strings differ by 17 bits, but does not consider whether this difference is between two objects of size 50 or 500. For this reason, the information distance is divided by the size of the largest description of the two strings, resulting in the normalized information distance (NID) described as:(3)$$NID\left(x,y\right)=\frac{\max\{K(x|y),K(y|x)\}}{\max\{K(x),K(y)\}}.$$

With the NID, one can measure the similarity between any string and, in particular, between any computer file. However, since this distance is calculated based on the representation of each object with the least Kolmogorov complexity, and this notion, in turn, is not computable, this measure is also not computable.

Two key ingredients are necessary to make the above information distance easy to implement in real computers. One is the manipulation of the expression to approximate max{K(x|y),K(y|x)} by K(xy)−min{K(x),K(y)}, where xy is the concatenation of *x* with *y*, using a Kolmogorov complexity version of symmetry of information. The second one is to use real-world compressors *Z* (e.g., *gzip*, *bzlib*, *zlib*) to approximate *K*, i.e., to use the output size of the real-world compressor to represent the Kolmogorov size of compressed objects. Having this in mind, the NID was rewritten to apply in the real world to the normalized compression distance (NCD) [25]:(4)$$NCD\left(x,y\right)=\frac{Z(xy)-\min\{Z(x),Z(y)\}}{\max\{Z(x),Z(y)\}}.$$

In practice, the NCD distance is calculated by first appending both files together and compressing them. Then, both files are compressed individually, taking the difference between the appended compressed file size and the most petite individual compressed file size. Finally, this difference is normalized by dividing it by the size of the most extensive individual compressed file. Since we are using compressors *Z* to approximate *K* of each object, it is clear that the better the *Z* is, the more accurate the NCD results will be.

### 2.3. The Quartet Method

So, the implementations of the quartet method described below use greedy algorithms to have an approximation to it. We describe the quartet method as, from our collection of *n* elements, we consider each group of four items. Therefore, the total number of groups is n4. A tree is then built from each group {u,v,w,x}, and each internal node has three neighbors, implying that the tree comprises two subtrees with two leaves each. Let this tree be called a quartet. There are three possible dispositions for a quartet: uv|wx, uw|vx, and ux|vw, where each pair of letters represents the leaves connected to the same internal node as shown in Figure 1.

The cost of each quartet is then defined by the sum of distances between each pair of neighbors: $C_{uv|wx}=d\left(u,v\right)+d\left(w,x\right)$. We say that a tree *T* is consistent with {u,v,w,x} if and only if the path from *u* to *v* does not cross the path from *w* to *x* for any given *T* and any collection of four leaf labels {u,v,w,x}. The tree representing our data can then be considered composed of this smaller quartet tree. Finally, the sum of the costs of all consistent quartets is the cost of the entire tree.

The authors of [16] go even further and implement a score *S* to measure how well any given tree represents the pairwise distance matrix. The score of a tree *T* of *N* leaves is computed as $S\left(T\right)=\left(M-C_{T}\right)/\left(M-m\right)$, a normalization of the cost of the tree *T*. The real value of the cost of a tree is given by $C_{T}=\sum_{u,v,w,x\subset N}\left\{C_{uv|wx}:T\;\mathrm{is}\;\mathrm{consistent}\;\mathrm{with}\;uv|wx\right\}$, i.e., it is the sum of the costs of all its consistent quartet topologies and the values *m* and *M* correspond to the best (minimal) cost $m\left(u,v,w,x\right)=\min\left\{C_{uv|wx},C_{uw|vx},C_{ux|vw}\right\}$, and a worst (maximal) cost $M\left(u,v,w,x\right)=\max\left\{C_{uv|wx},C_{uw|vx},C_{ux|vw}\right\}$, respectively. The higher the score, the lower the cost of the tree.

The problem of finding the best tree to represent the distance of the data is known to be NP-complete. Therefore, a method based on randomization and hill-climbing is used to find an approximation to the optimal tree. The search method starts by creating a random tree $T_{1}$ with n−2 internal nodes and *n* leaf nodes (2n−2 nodes total). The score $S(T_{1})$ is then calculated, and since it is the first score calculated, it denotes the best-known score up until that moment. A second tree $T_{2}$ is then created by mutating $T_{1}$ in three possible ways: (1) a leaf swap, choosing two leaf nodes randomly, and swapping them; (2) a subtree swap, choosing at random two internal nodes and swapping the subtrees rooted on them; and (3) a subtree transfer, choosing a random tree at random, detaching and reattaching it in another place, maintaining similarity invariants. After $T_{2}$ is created, we calculate its score $S(T_{2})$. If $S(T_{2})$ > $S(T_{1})$, then we keep the tree $T_{2}$; otherwise, we keep $T_{1}$ and repeat the process with a different configuration and choice. Finally, the search algorithm halts if it reaches a tree *T* with a score S(T)=1 or if no better trees are being found in a reasonable amount of time.

## 3. Complearn

Complearn [26] is a Linux command line tool that enables its users to perform “clustering by compression” straightforward and intuitively by providing all the resources necessary for distance matrix computation, cluster production, and visualization.

The tool is divided into two main command names, *ncd*, for computations of the Normalized Compression Distance and *maketree*, for binary tree generation from a given distance matrix based on hill-climbing algorithms, as described above.

The NCD command name offers the possibility to select between different compression methods among *bzlib*, *zlib*, and *blocksort*, and multiple input options described below:File Mode—takes, as an argument, a filename whose contents will be compressed.String Literal Mode—takes, as an argument, a string whose contents will be compressed. By default, each string literal is separated by white space. If a string contains literals with white space, that is surrounded with double quotes.Plain List Mode—takes, as an argument, a filename, which contains a list of filenames to be individually compressed. A line break separates each filename.Term List Mode—takes, as an argument, a filename whose contents are a list of string literals to be individually compressed. A line break separates each string character.Directory Mode—takes, as an argument, the name of a directory whose file contents will be used to compute the distance matrix.

The outputs from the *ncd* commands, if we so choose, can then be used as input for the *maketree* command. This command takes a distance matrix as input, computes an approximation of a possible best-fitting unrooted binary tree, and outputs a “treefile.dot” that can be visualized using the GraphViz tool [27] with the only two restrictions being that the matrix must be square and be of a size at least 4×4. Internally, the *maketree* method generates and represents its structure using the quartet method explained in Section 2.3.

## 4. Zgli

As mentioned as work objectives in Section 1, Zgli [21] is the tool we propose to perform clustering by compression using Python. It was developed in Python to ease its integration with other well-established data science utilities and enable more developers to use an NCD approach to their problems. This tool is divided into two main classes:Folder—this class performs operations inside a folder containing the files intended for compression and clustering.Encoder—this class was designed to perform all the operations regarding the tabular encoding of data.

In the subsequent subsections, we present the *Folder* and *Encoder* classes, explaining their purpose and main features. We will not explain in full detail all the functions for each of these classes, skipping over some details like function parameters, and some examples of use. For a detailed description of all the functions, code, usage, and a quick start guide, we refer the reader to the Zgli website [21].

### 4.1. Folder

Four separated functions compose the *Folder* class, three of which provide the user with access to relevant information about the files and one of which enables the user to generate a distance matrix that encompasses all the NCDs between the files in the folder.

The first three functions are *get_file_names*, *get_file_lengths*, and *get_file_sizes*, where length denotes the text file’s character count and size denotes the amount of memory it uses to be locally stored.

The final function in this class is *distance_matrix* function. Its purpose is to compute the NCD between all files inside the folder and give suitable compression approaches for specific characteristics the data may present.

Starting with compression options, Zgli has *gzip*, *lzma*, and *raw* compression options in addition to the pre-existing *zlib* and *bzlib* compressors already inbuilt in Complearn. The option *raw* means that the files were analyzed with the trivial compressor, i.e., with the identity. Therefore, with this option, the original file sizes are used in the computation of the NCD distance matrix.

The option to compress a file per column was the second feature we added to this function. This function will calculate the file size as the sum of the compressed sizes of all the columns inside the text file, if such a tabular structure exists. The user must provide the column delimiter (e.g., commas in CSV files).

Additionally, the user can perform weighted compression by column, where the compressed size of each column is multiplied by the weight provided, allowing the user to manipulate column importance on the final compressed file size.

The compression size CbC of the file using the option of compression by column of a file *x* given in tabular format with *n* columns is given as follows:(5)$$CbC\left(x\right)=\sum_{i=1}^{n}w_{i}\cdot Z\left(x_{i}\right)$$
where $w_{i}\in \left[0,1\right]$ is the given weight to each column *i* and *Z* is again the used compressor.

It is easy to verify that the weighted compression by columns respects the properties required to be classified as a normalized compression distance.

### 4.2. Encoder

The *Encoder* class has three different functions that allow the user to encode small tabular data into larger sequences, with the idea that the text file compressors are then capable of analyzing and compressing. This class was idealized due to the necessity to solve the problem that appears when compressing small text files using (almost optimal) state-of-the-art compression algorithms. Small files, even if entirely different, when compressed, will yield very similar compressed sizes, making the algorithm underlying the NCD unable to depict the differences between them correctly.

With the three encoding functions, *categorize_cols*, *standardize_categorical_cols*, and *encode_df*, provided in this class, the user can create a pipeline and encode the smaller data that is usually present inside the cells of tabular data, and still be able to use the clustering by compression approach to the data. In the following paragraphs, we discuss in detail an example of the use of this class. We use data taken from the iris plant dataset from the UCI Repository of Machine Learning Databases and Domain Theories [28]. The dataset contains three classes of 50 instances each, where each class refers to a type of iris plant. For this example, we will consider only 5 of the 150 instances, as shown in Table 1.

After loading the data, we encode all the columns comprised of continuous variables using the *categorize_cols* function. This function receives as an argument the number of categories to divide the column into, and outputs the table with each column having continuous values now assigned to category. Table 2 illustrates the use of this function when applied to the data present in Table 1, where we can see that every continuous value in first table is now represented by a range of values i.e., a category.

After using the *categorize_cols* function, all the data in our dataset should be categorical. All that is left to do before encoding is to make sure all columns have a standard representation that we can pass to the final *encode_df*. To accomplish this, we developed the *standardize_categorical_cols*, that maps every unique instance in a column to an ordinal sequence. This way we guarantee that every categorical column has the same representation format, even if there were categorical columns in the data other than the ones generated by the *categorize_cols*. Table 3 shows how data from Table 2 would look after using the *standardize_categorical_cols*.

Finally, with all data categorized and standardized, we can encode the features into patterned sequences using the *encode_df* function. Table 4 shows the final encoded version of our iris dataset slice.

The classes’ representations are created during the final and most significant encoding stage. Note that this task requires that the representations be so that compressors can detect differences between the classes while keeping a consistent distance between data of different classes.

To better understand how the *encode_df* function converts the data into categories, consider the following example, where we follow the categorization of a row from a dataset where the data was already categorized using *categorize_cols* and standardized using *standardize_categorical_cols*, similar to the data shown in Table 3:Row—0,1,0,2;ASCII string—0123456789abcdefg (...);Hop—1.
where the ASCII string is a string composed of 94 different ASCII characters, and the Hop is a user-defined parameter that determines how many characters should be jumped between different categories. The purpose of the Hop parameter will be clearer in due course at the end of the example.

The row is divided into classes 0, 1, and 2. This means that the encoding function will need to generate three distinct patterns to represent each one of these classes. All these patterns must have varying levels of complexity so that different compressed sizes are obtained. This variation in complexity must also be consistent across classes, so that feature importance remains generally constant. The solution proposed by *encode_df* for all of these issues is to encode the classes as follows:0—000000;1—010101;2—012012.

Every class is represented by a slice of the main ASCII string, with the size of this slice increasing by one (hop = 1) for each new class. For class 0, the slice is ’0’; for class 1, the slice is ’01’; for class 2, the slice is ’012’. If the value of the Hop parameter were equal to two, the size of this slice would increase by two for each new class, and the final encoding results would be:0—000000000000000;1—012012012012012;2—012340123401234.
where class 0 uses slice ’0’; class 1 uses slice ’012’; and class 2 uses slice ’01234’.

Additionally, an important aspect that one can observe in the previous example is that the size of the pattern string that represents each class grows proportionally with the number of classes and hop size. As previously stated, the typical behavior of a standard state-of-the-art compressor is for larger strings to generate larger compressed sizes. For this reason, all classes must have the same size strings representing them, so the differences between them rely on their complexity rather than their size. If we want all the strings to be of the same length, we must consider the Hop parameter and the number of classes the function must represent. Therefore, the size formula is given by:(6)size=lcm(hop,hop×2,⋯,hop×n)
where *n* is the number of column classes with the most classes in the data frame, and lcm is the least common multiple among all the numbers given as parameters. Furthermore, to ensure all the columns are represented in the same manner, this function ensures to distribute class’s representations between all the columns in the best way possible. For example, consider two columns *X*, taking the class values 0, 1 and 2, and *Y*, taking only the binary class values 0 and 1. Assume also that *X* is the column with the most classes in the dataset. A possible representation of its values, in the case with the initial hop of 2, could be the following:0—000000000000000;1—012012012012012;2—012340123401234.

To ensure that the binary values of column *Y* have the same impact on the compression sizes as the values of column *X*, the function encodes the two values of column *Y* as:0—000000000000000;1—012340123401234.
I.e., with the two values being represented with the same pattern as the two most distant values in the column *X*. This function considers the difference between class 0 and class 2 in column *X* to be the same as the difference between class 0 and class 1 in column *Y*, so it represents them in the same way. When this is not possible, the function rounds the class representation to the closest existing one in the column with the most features inside the data frame. When the approximation is tied, the value is rounded downwards.

## 5. Validation Tests

In this section, we investigate the new features implemented in the Zgli module. We design validation tests to understand whether the results obtained with new features perform better when compared with the previous features available in Complearn. These tests aim to verify if the Zgli tool enhances the state-of-the-art tools for clustering by compression in any manner other than to be added as a practical module for Python, especially integrated into packages for machine learning and data mining.

The main relevant research questions that these tests answer are:**Question 1:** Is it possible to improve the clustering by compression results of tabular data with the new compression by column option?**Question 2:** Is it possible to enhance the standard results for clustering (using the Euclidean distance, for example) for clustering categorical data using the Zgli encoder and the NCD?

### 5.1. Question 1—Improving Clustering Results Using Compressing by Columns

To answer the first research question, we have used a dataset created by David Guarin *et al.* and available at the UCI repository [29]. The dataset contains information of basketball participants performing different actions. An accelerometer (x,y,z) and a gyroscope (R, phi, delta) located on the player’s right arm were used to collect the data of movements. Four distinct users were requested to carry out several basketball moves: dribble, hold, pass, pickup, and shoot.

To circumvent the problem of the files being differentiated due to their different size, a separate dataset was built from the raw one (after the headers were removed) in which each file had the same number of lines. To have a reasonable number of line representatives, the same number of lines per file was accomplished by looping the smaller files and repeating them line by line until all the smaller files were the same size as the largest one—see Figure 2. Note that in this case, we are, on the one hand, not influencing the results dramatically, as the movements are expected to be repetitive. From the Kolmogorov complexity point of view, the information is the same as the smallest information in the initial pattern. Furthermore, in fact, to validate this method’s results, it was essential to observe that files will be clustered together based on their compressed size. Although not always, with state-of-the-art compressors, files with larger original sizes tend to have larger compressed sizes, and files with smaller original sizes tend to have smaller compressed sizes. Therefore, to have a term of comparison, we examine clusters formed using the actual file sizes, so we can later compare them to clusters generated using file compressors and see their impact on the groups created.

After looping all the necessary files in the 80-file dataset, we sampled 15 files (3 files per basketball move) just so the binary trees resulting from the clustering performed could be better visualized for the sake of explanation. Figure 3 shows the binary tree generated when using the uncompressed file sizes, and it clearly shows that the method is not able to correctly group the actions based on the raw size alone.

Now when looking at the results in Table 5 we can see that for both the raw dataset and the looped dataset where all files are of the same length, the tree scores (as defined in Section 3) are higher when the clusters are obtained using the compression by column option. Note that with a three-column dataset, the results improved approximately 2% for all the compressors considered in the tests.

The improvement of the results can also be seen in the binary tree image generated for the compressed by column option presented in Figure 4. When compared with Figure 5, for example the latter does not cluster properly dribbles and pickups, i.e., it does not correctly separate them by grouping them all in one branch of the tree, while in Figure 5, these two actions are now in distinct branches.

### 5.2. Question 2—Improving Results by Using Zgli Encoder

In this research question, we aim to see to which degree the encoder defined in Zgli enables the usage of an NCD approach to a problem comprised of small files that usually yield very similar compressed sizes. To answer this research question, we will use the iris dataset [28]. This dataset is composed of 3 classes of 50 instances each, where each class refers to a type of iris plant. One class is linearly separable from the other 2; the latter are not linearly separable from each other. We generated six different datasets from this original dataset, where a single file represents every row. We have files without any encoding for the original dataset, and for the remaining five, we have files encoded with the Zgli encoder using hop values from one to five.

The results in Table 6 show the accuracy scores of agglomerative clustering models generated with the distance matrices outputted by Zgli for each of these six datasets. We can observe the highlighted values in the table and verify that for each of the models, with the exception of the ones generated using the lzma compression algorithm, the Zgli encoder improved model results by at least 13% over using the results obtained using the original files without any encoding.

Table 6’s overall low accuracy scores shows the clustering by compression’s limitations when working with small files. Even so, the patterns generated by the hop encoding enable clustering by compression to produce a few outlier models with more respectable scores. For this reason, the hop encoder enhances the potential that pattern-generating algorithms can have when used as a step for clustering by compression. The exploration of such algorithms can lead to better models if more synergistic algorithms are found.

Finally, albeit briefly, the test findings in Table 6 highlight some intriguing qualities about available compressor possibilities and potential compressor choices. First, we observe that bzlib performed best with encoded sequences and generated a noticeably superior model when using encoded files as opposed to non-encoded data. When examining the best model created using the encoded files, Zlib also saw an increase in accuracy score, though not to the same level as the previous option. lzma was the only method that did not experience improvements in accuracy scores when employing encoded files. These factors lead us to believe that bzlib and zlib are probably more suitable for the zgli encoder module and agglomerative clustering than the lzma compression technique.

## 6. Clinical Use Case

Ankylosing spondylitis [30] (AS) is an autoimmune inflammatory condition belonging to the spondyloarthropathy category of rheumatic diseases. Its main early symptoms are back pain and early morning stiffness, and when left unchecked, these pains can extend to the whole spine and sacroiliac joints, with severe cases ending with the total fusion between the joints and bone structures in these areas. Most patients are young workers, and AS can result in significant socioeconomic hardship because patients must take time off work and, in severe situations, may have to quit their job.

With the advent of electronic medical records, the collected data provide insightful knowledge toward developing automatic decision helpers. In this use case, we generate clustering models using our Zgli tool to perform pattern mining on the Reuma.pt dataset [31] comprising clinical data from Portuguese patients with AS, and compare the results obtained with standard clustering approaches.

The patients’ data in this dataset are described with typical characteristics such as age and gender, but also include disease-specific features such as Bath Ankylosing Spondylitis Disease Activity Index (BASDAI) [32] and the more recent Ankylosing Spondylitis Disease Activity Score (ASDAS) [33], which correspond to indices that describe the disease’s activity and the level of impact it has on the patient. We refer the reader to [8] for a more detailed description of the data and the description of each disease index activity.

For this particular disease, doctors can prescribe four main biological treatments to AS patients: Etanercept, Infliximab, Golimumab, and Adalimumab. The ultimate goal of this dataset and the new techniques we have applied here is to find which treatment is the best for each patient based on his describing features. In this work, we analyzed treatment instances of patients with high or very high disease activity since these treatments aim to reduce this activity to low or inactive levels. For this reason, we only use treatment instances where the patient has a starting ASDAS of at least 2.1, as this is the accepted threshold for high disease activity [34]. Furthermore, we define treatment success and failure based on the ASAS-EULAR recommendations [35], with decreases of at least 1.1 on the starting ASDAS after 12 weeks being considered as successes and decreases lower than 1.1 being considered failures.

Now, we built a pipeline to analyze each dataset to find patterns between patients and treatments. This pipeline has Adalimumab, Etanercept, Golimumab, and Infliximab as starting points. We use zlib and bzlib as compression options, with lzma being left out based on the results from Section 5.2, with the added benefit of saving on the computational time it took to compress all the files using this algorithm. We then perform feature selection for each dataset using the Maximum Relevance Minimum Redundancy (MRMR) algorithm [36]. After obtaining the 10 best features for each dataset, we generate models starting by using the first two features with the best MRMR score, and appending the next best feature to the set until all the features were used to generate models. For each subset of features (2 best features, 3 best features, and so on until 10 best features), we also went over different numbers of clusters, from 2 to 10. Furthermore, for each pair of features and number of clusters, we used Hierarchical Clustering (with complete, average, and single distance) and K-medoids to generate 4 different models. Finally, when doing clustering by compression, we used the 2 fastest compression algorithms, *zlib* and *bzlib*, with both standard compression and compression by column. This means that for each dataset, we generate 1620 different models:9(featuresets)×9(clusters)×4(models)×5(typesofclustering)=1620
with feature sets being 9 because we went from a feature set of size 2 up to a feature set of size 10, clusters being 9 because we went from 2 clusters up to 10 clusters, models being 4 because we used hierarchical clustering with 3 different distances and K-medoids, and types of clustering being 5 because we used *bzlib* and *zlib* with standard compression and compression by column, and normal clustering.

These models were evaluated using three different scores: the silhouette score, v-measure, and adjusted random score [37,38,39]. The best-performing models for each of the scores were then analyzed, and finally, two main patterns between clusters and treatment were identified.

To evaluate the performance of the clustering by compression using Zgli when compared with standard clustering techniques, we produced tables with the top 10 models for each score. For clarity purposes, we will only look at the Golimumab results, show in Table 7, Table 8 and Table 9.

When analyzing the results, we see firstly that, when looking into clustering by compression in isolation, models with the highest values across all scores use the compression by column option, leading us to believe that this new feature is a good addition to the ones already present in Complearn.

Secondly, comparing clustering by compression approaches with conventional clustering approaches, the latter consistently beats the former when examining the scores alone. Despite this, clustering by compression demonstrated the ability to create models with competing outcomes while utilizing fewer clusters, indicating that the models required less population separation to produce comparable outcomes.

These clusters offered a new perspective on our clinical data and confirmed the general pattern that the patients could be neatly divided into two to four groups, with the categories being determined by the patient’s gender and initial ASDAS levels. The major difference between conventional and compression models was that the population was further divided into the different subsets of initial ASDAS values.

Patterns from both clustering approaches were extracted by looking at the feature distribution of the clusters of the best models. These clusters showed that treatment response failed mainly when ASDAS values were close to low disease activity (2.1 to 2.7 threshold) and were successful when ASDAS values were high to very high disease activity values (3.1 upwards). Treatment response in the interval from 2.7 to 3.1 was less predictable. Still, gender showed to be correlated to this response, with Infliximab, Golimumab, and Etanercept treatments having a higher success rate for male patients and a higher failure rate for female patients. The Adalimumab treatment was the sole exception to this rule, having similar success rates for both male and female patients.

## 7. Conclusions

The Zgli module proved to be more than simply a port of a similar tool into Python. All the new features implemented into the module are added value to the Complearn tool for users who intend to use clustering by compression as an approach to their problems. The method of compression by column had shown to be a way to improve clustering by compression results when dealing with tabular data, and the feature encoder enabled the technique to be employed with files that would otherwise yield compressed sizes differences to be too small to be used with this approach. Additionally, the tool is available in Python, making it easier to integrate clustering by compression with other popular machine-learning tools.

The Zgli tool also shows potential for its usage in the mining of data for significant patterns in real world problems such as those present in Reuma.pt. The resulting patterns from AS patients show that the Infliximab, Golimumab, and Etanercept treatments have a higher success rate for male patients and a higher failure rate for female patients, with the Adalimumab treatment being the only exception, with similar success rates for both male and female patients. We could also see a direct relation between initial ASDAS and our target variable where patients with higher ASDAS values had higher success rates and vice-versa. This use case ultimately showed that clustering by compression can compete with standard clustering and that the treatment choice for AS could follow a logic found in the patterns uncovered from patients’ data.

## Figures and Tables

**Figure 1 sensors-23-01219-f001:**
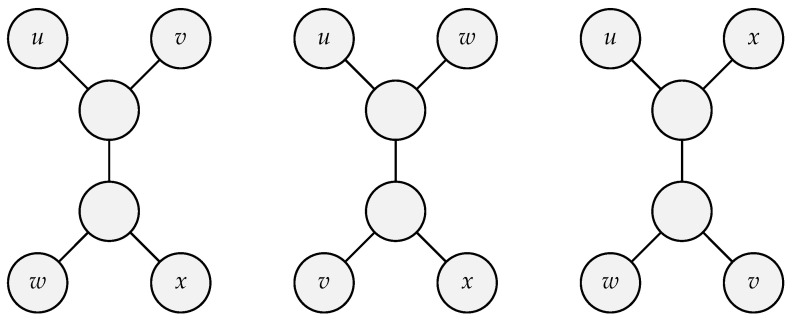
All possible position of quartets of four nodes.

**Figure 2 sensors-23-01219-f002:**
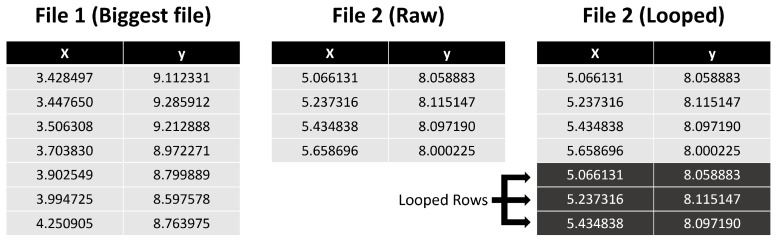
Looped rows example.

**Figure 3 sensors-23-01219-f003:**
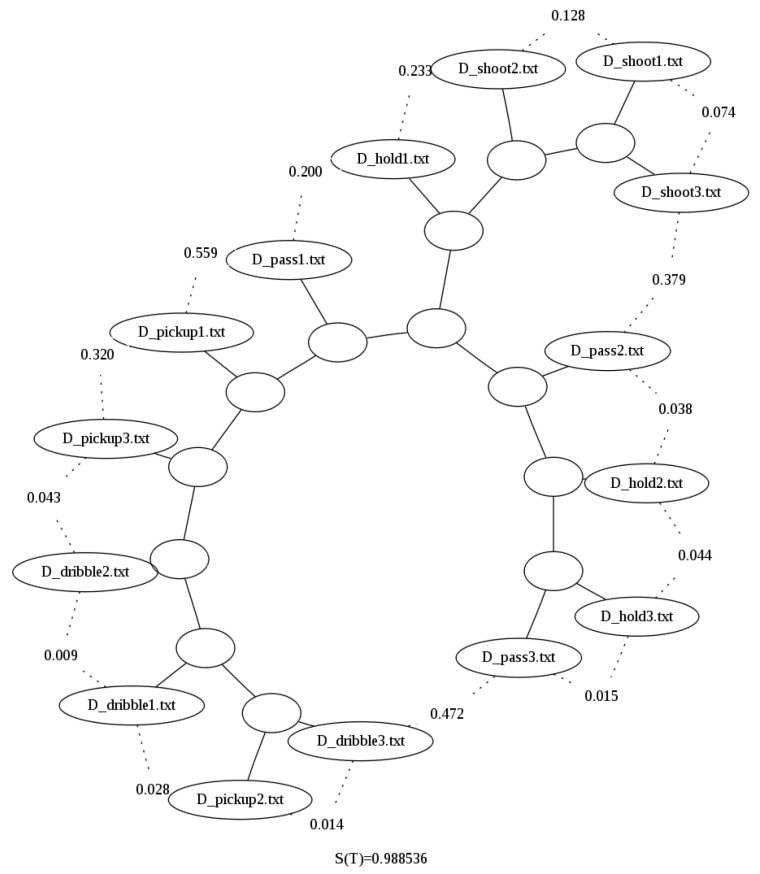
Binary tree generated using Zgli on the raw sizes of the Basketball data [29]. It can be seen that the shootings are the only type of data that is clustered together. Furthermore, the rest of the data is clustered linearly.

**Figure 4 sensors-23-01219-f004:**
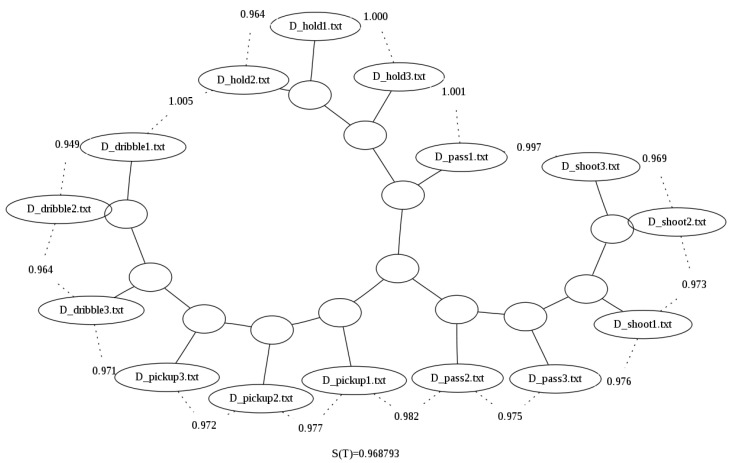
Binary tree generated using the looped data and bzlib without compression by column over the Basketball data set [29].

**Figure 5 sensors-23-01219-f005:**
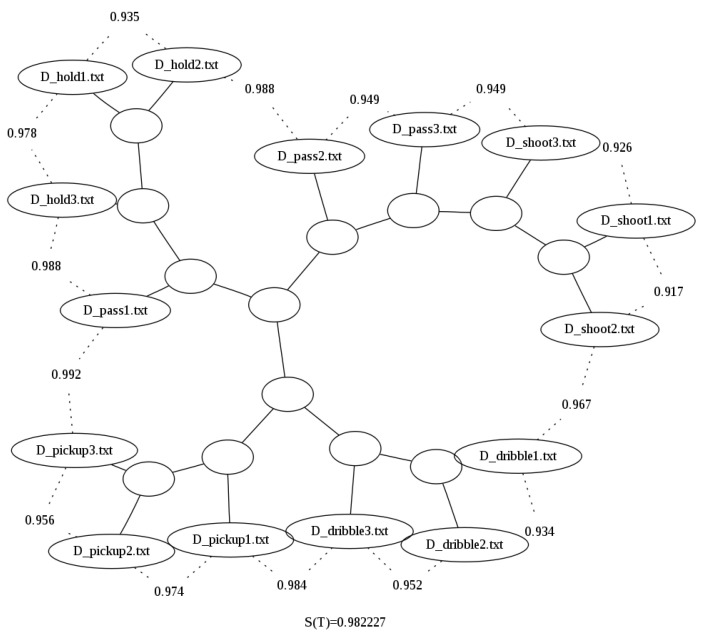
Binary tree generated using the looped data and bzlib with compression by column over the Basketball data set [29].

**Table 1 sensors-23-01219-t001:** Raw tabular data example extracted from iris plant dataset [28].

Feature1	Feature2	Feature3	Feature4
5.1	3.5	1.4	0.2
4.9	3.0	1.4	0.2
7.0	3.2	4.7	1.4
6.4	3.2	4.5	1.5
6.3	3.3	6.0	2.5 0.2

**Table 2 sensors-23-01219-t002:** Categorized data obtained after applying *categorize_cols* to the raw data example extracted from the iris data set presented in Table 1.

Feature1	Feature2	Feature3	Feature4
(4.296, 5.2]	(3.2, 3.8]	(0.994, 2.475]	(0.0976, 0.7]
(4.296, 5.2]	(2.6, 3.2]	(0.994, 2.475]	(0.0976, 0.7]
(6.1, 7.0]	(2.6, 3.2]	(3.95, 5.425]	(1.3, 1.9]
(6.1, 7.0]	(2.6, 3.2]	(3.95, 5.425]	(1.3, 1.9]
(6.1, 7.0]	(3.2, 3.8]	(5.425, 6.9]	(1.9, 2.5]

**Table 3 sensors-23-01219-t003:** Standardized data obtained after applying the *standardize_categorical_cols* to Table 2.

Feature1	Feature2	Feature3	Feature4
0	2	0	0
0	1	0	0
2	1	2	2
2	1	2	2
2	2	3	3

**Table 4 sensors-23-01219-t004:** Encoded data obtained after applying *encode_df* to the standardized data of Table 3.

Feature1	Feature2	Feature3	Feature4
000000000000	012012012012	000000000000	000000000000
000000000000	010101010101	000000000000	000000000000
012012012012	010101010101	012012012012	012012012012
012012012012	010101010101	012012012012	012012012012
012012012012	012012012012	012301230123	012301230123

**Table 5 sensors-23-01219-t005:** Comparison of tree scores obtained using two types of datasets: raw and looped dataset, with and without the option of compression by column being active. The compressors used are the ones inbuilt into the original Complearn tool [16].

Compressor	Dataset	Compress by Column Option	Tree Scores
bzlib	raw	Disabled	0.968793
bzlib	raw	Enabled	0.982927
bzlib	looped	Disabled	0.968793
bzlib	looped	Enabled	0.982227
zlib	raw	Disabled	0.979618
zlib	raw	Enabled	0.991362
zlib	looped	Disabled	0.979618
zlib	looped	Enabled	0.991362
lzma	raw	Disabled	0.996360
lzma	raw	Enabled	0.991362
lzma	looped	Disabled	0.976434
lzma	looped	Enabled	0.996360

**Table 6 sensors-23-01219-t006:** Accuracy scores of Agglomerative clustering models generated using NCD distance matrices. The boldface scores show the best score of each model. Agg Ave Acc—Accuracy of agglomerative clustering using average linkage; Agg Com Acc—Accuracy of agglomerative clustering using complete linkage; and Agg Sin Acc—Accuracy of agglomerative clustering using single linkage.

Compressor	Hop	Agg Ave Acc	Agg Com Acc	Agg Sin Acc
bzlib	1	0.133	0.333	0.200
bzlib	2	0.533	0.200	0.400
bzlib	3	**0.733**	0.333	**0.733**
bzlib	4	0.733	0.067	0.200
bzlib	5	0.400	0.133	0.400
bzlib	normal	0.000	0.133	0.400
lzma	1	0.333	0.067	0.333
lzma	2	0.400	**0.600**	0.333
lzma	3	0.067	0.133	0.400
lzma	4	0.533	0.267	0.200
lzma	5	0.533	0.067	0.133
lzma	normal	0.600	0.400	0.400
zlib	1	0.600	0.267	0.200
zlib	2	0.067	0.133	0.467
zlib	3	0.400	0.133	0.267
zlib	4	0.200	0.133	0.400
zlib	5	0.400	0.133	0.467
zlib	normal	0.467	0.200	0.333

**Table 7 sensors-23-01219-t007:** Silhouette scores of the best five clustering by compression and the best five standard compression models for the Golimumab dataset. N/A means that no compressor was applied.

Score	Clusters	Features	Compressor	Model
0.588178	8	2	N/A	HC complete
0.583808	8	2	N/A	HC average
0.56795	2	3	bzlib by column	HC average
0.56795	2	3	bzlib by column	K-medoids
0.56795	2	3	bzlib by column	HC complete
0.56795	2	3	bzlib by column	HC single
0.505713	8	4	bzlib by column	HC complete
0.475152	8	4	N/A	HC complete
0.446612	7	4	N/A	HC average
0.441733	6	4	N/A	HC single

**Table 8 sensors-23-01219-t008:** V-measures of the best five clustering by compression and the best five standard compression models for the Golimumab dataset. N/A means that no compressor was applied.

Score	Clusters	Features	Compressor	Model
0.588178	8	2	N/A	HC complete
0.505713	8	4	bzlib by column	HC complete
0.502198	8	4	bzlib by column	HC single
0.475152	8	4	N/A	HC complete
0.450261	8	5	bzlib by column	HC complete
0.446612	7	4	N/A	HC average
0.425628	8	5	N/A	HC complete
0.390852	3	3	bzlib by column	K-medoids
0.390682	8	5	bzlib by column	HC single
0.386077	8	5	N/A	HC average

**Table 9 sensors-23-01219-t009:** Adjusted random scores of the best five clustering by compression and the best five standard compression models for the Golimumab dataset. N/A means that no compressor was applied.

Score	Clusters	Features	Compressor	Model
0.446612	7	4	N/A	HC complete
0.441733	6	4	N/A	HC single
0.432458	8	4	N/A	HC average
0.412846	3	3	bzlib by column	HC complete
0.407364	7	5	N/A	HC complete
0.390852	3	3	bzlib by column	K-medoids
0.390682	8	5	bzlib by column	HC single
0.386953	5	5	N/A	HC average
0.374891	3	5	bzlib by column	HC complete
0.308163	3	3	zlib by column	HC average

## Data Availability

The Iris and Basketball sets used in this paper can be obtained from the following link: https://archive.ics.uci.edu/ml/datasets/iris and https://archive.ics.uci.edu/ml/datasets/Basketball+dataset accessed on 18 January 2023, respectively. Data from Reuma.pt are not publicly available.

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
