# Peer review of "Zgli: A Pipeline for Clustering by Compression with Application to Patient Stratification in Spondyloarthritis"

_sensors, 2023, doi:10.3390/s23031219_

Round 1

Reviewer 1 Report

In their article "Zgli: A pipeline for clustering by compression with application to patient stratification in Spondyloarthritis" the authors present a new approach to clustering by compression that extends the existing Linux Complearn tool. Their approach has several advantages compared to Complearn, for example it is written in Python and provides a novel compression-by-column feature. The authors explain their approach in a clear way and convincingly argue that Zgli has an advantage over Complearn. They also apply their approach to an interesting real-world medical use case.

This article contains new algorithms and results that are interesting to the community working on clustering and compression. In my opinion the applications are chosen nicely. Overall I enjoyed reading the article very much. Also, the Zgli website (their reference [21]) represents a nice addition to the article. Therefore I am happy to recommend the publication of this article.

Author Response

Answer's to Reviewer 1 

{\em 
In their article "Zgli: A pipeline for clustering by compression with application to patient stratification in Spondyloarthritis" the authors present a new approach to clustering by compression that extends the existing Linux Complearn tool. Their approach has several advantages compared to Complearn, for example it is written in Python and provides a novel compression-by-column feature. The authors explain their approach in a clear way and convincingly argue that Zgli has an advantage over Complearn. They also apply their approach to an interesting real-world medical use case.

This article contains new algorithms and results that are interesting to the community working on clustering and compression. In my opinion the applications are chosen nicely. Overall I enjoyed reading the article very much. Also, the Zgli website (their reference [21]) represents a nice addition to the article. Therefore I am happy to recommend the publication of this article.}

\bf {Our answer:}

We thank the reviewer for the effort of reading our paper and the sympathy toward the publication of the paper. 

Reviewer 2 Report

1.       The authors developed a tool named zgli to extend existing tool CompLearn which uses NCD and quartet method of tree construction. The title can be misleading because there is the expectation that the contribution is on the clustering but it was found that it is applying an existing clustering by compression technique. Perhaps consider: “Zgli: A pipeline for applying clustering by compression with application to patient stratification in Spondyloarthritis”

2.       The work is helpful as the authors provided a library for practitioners to use clustering by compression on tabular data in different files in a folder

3.       Overall the paper is well-written and organized. We would like to seek further clarification from authors as follows:

a.       Is the tree scores computed based on cost of tree - sum of  cost of relevant quartets? Can this be made explicit?

b.       Table 5 - lzma produced highest tree score but performed poorly during clustering. Does this not mean that tree score is not a good evaluation metric for determining suitable compressor for clustering?

c.       Results are accuracy centric with little discussion or insights.

d.       We see very poor results in Table 6, some less than 0.5, what does this mean with respect to the hop or the compressor?

e.       Does this mean that lzma and zlib should not be considered  as compression methods to be used for clustering?

f.        Why lzma and zlib performed much poorer?

g.       Why average and single linkage perform better with compression but not average? Has this got to do with the tree construction?

h.       Table 6 - lzma performed better  with normal than using encoding with hops, assuming normal is without using encoding

i.        Table 6 shows HC with average producing the lowest acc. But in Table 7-9, HC Complete produced the best results without compression. Please explain this.

j.        From the results, we can see results improvement between standard clustering and clustering by compression approach; however, it is not clear whether the improvement is by the compression technique or the encoding technique?

k.       From Table 7, 8 and 9 evaluation measures; those that use  compressor had lower scores e.g Table 8 with standard clustering had v-measure of 0.588 and with compression 0.5057. Does not seem convincing that clustering by compression is effective

l.        There is not much insightful discussion regarding how was clustering by compression useful or better; as results are mainly overall performance.

m.      On specific treatments, it is unclear why "models show that the treatments Infliximab, Golimumab, and Etanercept have a greater success rate for male patients" - please discuss

Author Response

Answer's to Reviewer 2

1.       The authors developed a tool named zgli to extend existing tool CompLearn which uses NCD and quartet method of tree construction. The title can be misleading because there is the expectation that the contribution is on the clustering but it was found that it is applying an existing clustering by compression technique. Perhaps consider: “Zgli: A pipeline for applying clustering by compression with application to patient stratification in Spondyloarthritis”

2.       The work is helpful as the authors provided a library for practitioners to use clustering by compression on tabular data in different files in a folder

Our answer:

We thank the reviewer for the effort in reading our paper and the nice comments that helped us improve the presentation of the paper and our results. 

%%%%%%%%%%%%%%%%%%%%%%%%%%%%%%%%%%%%%%%%%%%%%%%%%%%%%%%%%%%%%%%%%%%%%%%%%%%%%%%%%%%%%%%%%%%%%%%%%%%%%%%%%%%%%%

3.       Overall the paper is well-written and organized. We would like to seek further clarification from authors as follows:

a.       Is the tree scores computed based on cost of tree - sum of  cost of relevant quartets? Can this be made explicit?

Our answer:

We thank the reviewer for the comment. The authors of the original idea of using compression for clustering [26] explained this point adequately. We reformulated the text to include a description of how a tree's score is computed by rewriting part of the paragraph starting in line 133 to address this comment in the text:

The score of a tree $T$ of $N$ leaves is computed as $(T) = (M − C_T )/(M − m)$, a normalization of the cost of the tree $T$. The real value of the cost of a tree is given by $C_T =
\sum_{{u,v,w,x}\subset N} \{C_{uv|wx} : T\textnormal{ is consistent with }uv|wx\}$, i.e., it is the sum of the costs of all its consistent quartet topologies and the values $m$ and $M$ correspond to the  best (minimal) cost $m(u, v, w, x) = \min \{C_{uv|wx}, C_{uw|vx}, C_{ux|vw}\}$, and a worst (maximal) cost
$M(u, v, w, x) = \max \{C_{uv|wx}, C_{uw|vx}, C_{ux|vw}\}$, respectively. The higher the score, the lower the cost of the tree.

%%%%%%%%%%%%%%%%%%%%%%%%%%%%%%%%%%%%%%%%%%%%%%%%%%%%%%%%%%%%%%%%%%%%%%%%%%%%%%%%%%%%%%%%%%%%%%%%%%%%%%%%%%%%%%

b.       Table 5 - lzma produced highest tree score but performed poorly during clustering. Does this not mean that tree score is not a good evaluation metric for determining suitable compressor for clustering?

Our answer:

We appreciate the reviewer's feedback. Because we did not adequately explain table 6 in its analysis, this question and a few others raised later in this assessment emerged. This issue was addressed in the updated version of the paper.

Regarding this particular issue, it is not that the lzma algorithm performs poorly at clustering; instead, it does not appear in our clinical data analysis when examining the clusters created. It does not occur since the algorithm was not applied; instead, we could confirm through Table 6's results that the lzma compressor's performance in model scores was not improved by including the encoder. Therefore, we decided to test the encoder with bzlib and zlib since they showed the potential to produce models with the most significant gains, saving the computational time the lzma technique required to compress all the files. To clarify this matter, we rewrite the text as follows, highlighting our decision in the sentence of the paragraph beginning on line 416:

"We use zlib and bzlib as compression options, with lzma being left out based on the results from Section 5.2, with the added benefit of saving on the computational time it took to compress all the files using this algorithm".

%%%%%%%%%%%%%%%%%%%%%%%%%%%%%%%%%%%%%%%%%%%%%%%%%%%%%%%%%%%%%%%%%%%%%%%%%%%%%%%%%%%%%%%%%%%%%%%%%%%%%%%%%%%%%%

c.       Results are accuracy centric with little discussion or insights.

d.       We see very poor results in Table 6, some less than 0.5, what does this mean with respect to the hop or the compressor?

Our answer to the last tow comments:

We agree with the reviewer that the results and discussion strongly emphasize accuracy. We expand on table 6's analysis by discussing the low score values and what they signify, as well as the conclusions we came to as a result of these findings. The sentence that begins on line 372 responds to the comments marked "c" and "d":

"Table~\ref{table:iris_acc}'s overall low accuracy scores is an example of clustering by compression's limitations when working with small files. Even so, the patterns generated by the hop encoding enable clustering by compression to produce a few outlier models with more respectable scores. For this reason, the hop encoder enhances the potential that pattern-generating algorithms can have when used as a step for clustering by compression. The exploration of such algorithms can lead to better models if more synergistic algorithms are found."

%%%%%%%%%%%%%%%%%%%%%%%%%%%%%%%%%%%%%%%%%%%%%%%%%%%%%%%%%%%%%%%%%%%%%%%%%%%%%%%%%%%%%%%%%%%%%%%%%%%%%%%%%%%%%%

e.       Does this mean that lzma and zlib should not be considered  as compression methods to be used for clustering?

Our answer:

We thank the reviewer for making this observation. We believe that lzma should not be used with our encoding technique based on the results we found and are displayed in table 6. Still, we do not entirely rule out the chance that it could produce superior models with a different type of encoder and data. Given the many different forms of data available for any given problem and the similar trend we saw while examining zlib, we do not completely rule out the chance that zlib may perform better than bzlib when combined with our zgli encoder solution. We added a new paragraph addressing this, starting in line 378:

"Finally,  albeit brief, the test findings in Table~\ref{table:iris_acc} highlight some intriguing qualities about available compressor possibilities and potential compressor choices. First, we observe that bzlib performed best with encoded sequences and generated a noticeably superior model when using encoded files as opposed to non-encoded data. When examining the best model created using the encoded files, Zlib also saw an increase in accuracy score, though not to the same level as the previous option.  lzma was the only method that did not experience improvements in accuracy scores when employing encoded files. These factors lead us to believe that bzlib and zlib are probably more suitable for the zgli encoder module and agglomerative clustering than the lzma compression technique."

%%%%%%%%%%%%%%%%%%%%%%%%%%%%%%%%%%%%%%%%%%%%%%%%%%%%%%%%%%%%%%%%%%%%%%%%%%%%%%%%%%%%%%%%%%%%%%%%%%%%%%%%%%%%%%

f.        Why lzma and zlib performed much poorer?

h.       Table 6 - lzma performed better  with normal than using encoding with hops, assuming normal is without using encoding

Our answer to the last two comments:

We believe that the structure and solution of the algorithm are the answer to these two questions. As we previously stated, the results from table 6 led us to reject employing the lzma algorithm for our clinical data as it is anticipated that using the lzma technique with hop encoding will not increase the performance of our models.

Both the lzma and zlib algorithms may produce inferior results because they are developed to compress long data sequences, and their solutions look for patterns while taking big sequential chunks of data into account. Contrarily, Bzlib often performs better with shorter data sequences because it employs a more linear search strategy for patterns. The results suggest that the hop encoding technique provides sequences more suitable for compression using the latter algorithm than the first two, in addition to these other factors.

%%%%%%%%%%%%%%%%%%%%%%%%%%%%%%%%%%%%%%%%%%%%%%%%%%%%%%%%%%%%%%%%%%%%%%%%%%%%%%%%%%%%%%%%%%%%%%%%%%%%%%%%%%%%%%

g.       Why average and single linkage perform better with compression but not complete? Has this got to do with the tree construction?

Our answer:

We appreciate the reviewer asking this question. The issue raised is an excellent example of how data may affect how models are constructed. Three of the four main types of distances employed in agglomerate clustering are compatible with already computed distance matrices (which is our case). These are single, average, and complete distances. The iris dataset's clustering findings are shown in table six. Because the classes are clear and straightforward to distinguish, the single and average distances fared better. On the other hand, models created using the complete distance showed to be superior when using our clinical data in situations where patient clusters are not as clearly defined. 

Distance performance is mainly related to the data, and it is hard to say which distance will generate the best models before generating them.

%%%%%%%%%%%%%%%%%%%%%%%%%%%%%%%%%%%%%%%%%%%%%%%%%%%%%%%%%%%%%%%%%%%%%%%%%%%%%%%%%%%%%%%%%%%%%%%%%%%%%%%%%%%%%%
i.        Table 6 shows HC with average producing the lowest acc. But in Table 7-9, HC Complete produced the best results without compression. Please explain this.

Our answer:

We thank the reviewer for this comment, but we believe that he/she may have misunderstood Table 6, since all models in the table are produced using the NCD, i.e., using compression.

Conventional models and clustering by compression models are the two kinds of models to consider, as we have already described in previous paragraphs. The first group usually beats the second, as shown in Tables 7-9. This is expectable given that the data employed synergizes very well with conventional models but not as well with compressors. These experiments determine how much model performance varies and whether applying clustering by compression models to the data results in any novel discoveries.

%%%%%%%%%%%%%%%%%%%%%%%%%%%%%%%%%%%%%%%%%%%%%%%%%%%%%%%%%%%%%%%%%%%%%%%%%%%%%%%%%%%%%%%%%%%%%%%%%%%%%%%%%%%%%%

j.        From the results, we can see results improvement between standard clustering and clustering by compression approach; however, it is not clear whether the improvement is by the compression technique or the encoding technique?

Our answer:

We thank the reviewer for this comment. As mentioned in other comments, we think that explanation of the tables may not have been clear. We polished the text to explain the results properly. Furthermore, and answering directly to the reviewer's question, we may say that using the Complearn tool, we were able to generate models with an accuracy of 0.466 for the iris dataset when using the best-performing compression option (zlib). What we want to show is that new implementations in the zgli tool, like lzma, as well as new techniques like data encoding, allow this technique to achieve higher accuracy values. As a result, we can conclude that both compression and encoding techniques affect the model results. We intend to shed light on these discoveries. We believe that additional research will result in even better models (with other types of compressors or other possible encoding techniques) for this unique clustering approach.

%%%%%%%%%%%%%%%%%%%%%%%%%%%%%%%%%%%%%%%%%%%%%%%%%%%%%%%%%%%%%%%%%%%%%%%%%%%%%%%%%%%%%%%%%%%%%%%%%%%%%%%%%%%%%%
k.       From Table 7, 8 and 9 evaluation measures; those that use  compressor had lower scores e.g Table 8 with standard clustering had v-measure of 0.588 and with compression 0.5057. Does not seem convincing that clustering by compression is effective.

Our answer:

We agree with the reviewer. Clustering by compression is not as effective as conventional clustering when looking at model scores. Even so, some clustering by compression models were able to obtain competing results when looking at other scores, such as the Silhouette score and the Adjusted Random Score, and do so by clustering the data using different features and cluster numbers. These models can represent a different way of looking at the data and can confirm patterns already uncovered by the convectional clustering models generated or even uncover new ones. We mention this in the paragraph staring starting in line 434:

"Secondly, comparing clustering by compression approaches with conventional clustering approaches, the latter consistently beats the former when examining the scores alone. Despite this, clustering by compression demonstrated the ability to create models with competing outcomes while utilizing fewer clusters, indicating that the models required less population separation to produce comparable outcomes. "

%%%%%%%%%%%%%%%%%%%%%%%%%%%%%%%%%%%%%%%%%%%%%%%%%%%%%%%%%%%%%%%%%%%%%%%%%%%%%%%%%%%%%%%%%%%%%%%%%%%%%%%%%%%%%%

l.        There is not much insightful discussion regarding how was clustering by compression useful or better; as results are mainly overall performance.

Our answer:

We thank the reviewer for pointing this out. As mentioned before, clustering by compression models can be helpful by confirming patterns uncovered by the better-scored convectional models, or even uncovering new ones, seeing that the data is grouped using different numbers of clusters and features from the conventional models. We discuss this in the reformulated paragraph starting in line 439:

"These clusters offered a new perspective on our clinical data and confirmed the general pattern that the patients could be neatly divided into two to four groups, with the categories being determined by the patient's gender and initial ASDAS levels. The major difference between conventional and compression models was that the population was further divided into the different subsets of initial ASDAS values. "

%%%%%%%%%%%%%%%%%%%%%%%%%%%%%%%%%%%%%%%%%%%%%%%%%%%%%%%%%%%%%%%%%%%%%%%%%%%%%%%%%%%%%%%%%%%%%%%%%%%%%%%%%%%%%%
m.      On specific treatments, it is unclear why "models show that the treatments Infliximab, Golimumab, and Etanercept have a greater success rate for male patients" - please discuss

Our answer:

We thank the reviewer for mentioning this. We further elaborate on these results in the new paragraph starting in line 434:

"Patterns from both clustering approaches were extracted by looking at the feature distribution of the clusters of the best models. These clusters showed us that treatment response failed mainly when ASDAS values were close to low disease activity (2.1 to 2.7 threshold) and were successful when ASDAS values were high to very high disease activity values (3.1 upwards). Treatment response in the interval from 2.7 to 3.1 was less predictable, but gender showed to be correlated to this response, with Infliximab, Golimumab, and Etanercept treatments having a higher success rate for male patients and a higher failure rate for female patients. The Adalimumab treatment was the sole exception to this rule, having similar success rates for both male and female patients."

Reviewer 3 Report

* Tables 1, 2,  3, &4 are not properly referenced in the text. I suggest explaining the dataset and its structure in the text and hence using shorter table titles. Consider this for all table titles.

* Can you explain why the encoded data shown in Table 4 is all zeros for three of the features? 

* Line 275: "this function ensures to distribute class’s representations between all the columns in the best way possible". I am afraid the example you included is not clear enough and hence very hard to follow.

* I found some difficulty following the example used for section 5. validation Test. I believe it needs to be restructured and/or rewritten. For example, Table 5 should show the improvement of compression by column. However, It is hard to reach a conclusion since the cases are grouped by "compressor" type rather than "Compress by Column" or not. Furthermore, I believe some discussion is needed on how the 2% improvement is considerable in this context. 

* The results listed in Table 6 need to be discussed a little further. For example, why the accuracy sometimes drops with higher hop steps? what are you highlighting by the boldface?

* The findings summarized in lines 391-397 are not directly related to the results in tables 7, 8, & 9. The authors described the process but didn't show how this was applied to the data. If this is too large to be included in the article due to space restrictions, please consider adding them as supplementary material.

Author Response

Answer's to Reviewer 3 

* Tables 1, 2,  3, &4 are not properly referenced in the text. I suggest explaining the dataset and its structure in  the text and hence using shorter table titles. Consider this for all table titles.

Our answer:

We appreciate the reviewer's feedback. We appreciate the reviewer's feedback. To more accurately describe the information in each table, we rewrote the text pertaining to the encoding pipeline. We also provide a brief and straightforward explanation of how each of the functions used in each table operates. After expanding the textual data explanation, we shortened the table titles. These changes can be seen in the paragraphs starting in lines 232, 236, 242, and 250. We also provide a brief and straightforward explanation of how each function used in each table operates. After expanding the textual data explanation, we shortened the table titles. These changes can be seen in the paragraphs starting in lines 232, 236, 242, and 250.

%%%%%%%%%%%%%%%%%%%%%%%%%%%%%%%%%%%%%%%%%%%%%%%%%%%%%%%%%%%%%%%%%%%%%%%%%%%%%%%%%%%%%%%%%%%%%%%%%%%%%%%%%%%%%%

* Can you explain why the encoded data shown in Table 4 is all zeros for three of the features? 

Our answer:

Each distinct category of the classes in our standard dataframe (shown in Table 2) is represented by a patterned sequence, as we mentioned in the explanation of the encoder function. Consequently, the standard dataframe's zeros will all be represented by the same sequence (000000000000). 

Because the standard data in Table 2 was likewise all zeros for three of the features, the encoded data given in Table 4 is also all zeros for those three features.

We think this no to be a good example reflective of the encoding function, and so, in order to avoid future readers asking the same question as the reviewer, we chose a different slice of the iris dataset that can offer a wider variety of categories and, thus, a broader range of sequences. Tables 3 and 4, in particular, where the categories and sequences are no longer primarily 0s, show these changes best, but it is important to note that for this to happen, data in Tables 1 and 2 also had to be changed.

%%%%%%%%%%%%%%%%%%%%%%%%%%%%%%%%%%%%%%%%%%%%%%%%%%%%%%%%%%%%%%%%%%%%%%%%%%%%%%%%%%%%%%%%%%%%%%%%%%%%%%%%%%%%%%
* Line 275: "this function ensures to distribute class’s representations between all the columns in the best way possible". I am afraid the example you included is not clear enough and hence very hard to follow.

Our answer:

We agree with the reviewer that the example could be better presented. We rewrote, starting on line 289, the text following that sentence to address the issue raised by the reviewer.  

{\color{blue} For sake of example consider two columns $X$, taking class values $0$, $1$ and $2$ and $Y$, taking only binary class values $0$ and $1$. Assume also that $X$ is the column with the most classes in the dataset. A possible representation of its values, in the case with the initial hop of $2$, could be the following:
\begin{itemize}
    \item 0 -- 000000000000000
    \item 1 -- 012012012012012
    \item 2 -- 012340123401234
\end{itemize}

To ensure that the binary values of column $Y$ have the same impact on the compression sizes as the values of column $X$, the function encodes the two values of column $Y$ as:
\begin{itemize}
    \item 0 -- 000000000000000
    \item 1 -- 012340123401234
\end{itemize}
i.e., with the two values being represented with the same pattern as the two most distant values in the column $X$.}

%%%%%%%%%%%%%%%%%%%%%%%%%%%%%%%%%%%%%%%%%%%%%%%%%%%%%%%%%%%%%%%%%%%%%%%%%%%%%%%%%%%%%%%%%%%%%%%%%%%%%%%%%%%%%%
* I found some difficulty following the example used for section 5. validation Test. I believe it needs to be restructured and/or rewritten. For example, Table 5 should show the improvement of compression by column. However, It is hard to reach a conclusion since the cases are grouped by "compressor" type rather than "Compress by Column" or not. Furthermore, I believe some discussion is needed on how the 2% improvement is considerable in this context. 

Our answer:

We thank the reviewer for this comment and the suggestions to improve the presentation of the results. We believe that the results presented in this section were hard to read for the reviewer due to two main issues: 1) the difficulty of the reviewer (according to previous comments) to understand the first part of the paper where we explain what our proposal is; 2) the discussion of the results.

Following the reviewer's first comment, we think the rewriting we made solves the first issue, and the discussion of the results we added in line 277 of the new version of the paper solves the second issue.

Regarding Table 5, we have a different perspective from the reviewer. We believe it is easier for the reader to evaluate the improvement of the results by using the compression by column option, to have for each compressor and each dataset (raw and looped) the results of having the "compression by column" active or not. To help even further the reader, we opt to relabel the column "compress by column" to "compress by column option" and its values from "true" and "false" to "enabled" and "disabled".  

Regarding the improvement of the results of 2\%, we agree with the reviewer and added the following text in line 349.  

{\color{blue} Note that with a three-column dataset, the results improved approximately $2\%$ for all the compressors considered in the tests.} 

%%%%%%%%%%%%%%%%%%%%%%%%%%%%%%%%%%%%%%%%%%%%%%%%%%%%%%%%%%%%%%%%%%%%%%%%%%%%%%%%%%%%%%%%%%%%%%%%%%%%%%%%%%%%%%
* The results listed in Table 6 need to be discussed a little further. For example, why the accuracy sometimes drops with higher hop steps? what are you highlighting by the boldface?

Our answer:

We thank the reviewer for bringing this to our attention and agree with this remark. Starting at line 372, we address the results in table 6 in great detail. Here, we discuss the outlier models marked in boldface in the table, explain the significance of the low accuracy scores for most models, and explore the insights that may be drawn from these findings.

"Table~\ref{table:iris_acc}'s overall low accuracy scores is an example of clustering by compression's limitations when working with small files. Even so, the patterns generated by the hop encoding enable clustering by compression to produce a few outlier models with more respectable scores. For this reason, the hop encoder enhances the potential that pattern-generating algorithms can have when used as a step for clustering by compression. The exploration of such algorithms can lead to better models if more synergistic algorithms are found.

Finally,  albeit brief, the test findings in Table~\ref{table:iris_acc} highlight some intriguing qualities about available compressor possibilities and potential compressor choices. First, we observe that bzlib performed best with encoded sequences and generated a noticeably superior model when using encoded files as opposed to non-encoded data. When examining the best model created using the encoded files, Zlib also saw an increase in accuracy score, though not to the same level as the previous option.  lzma was the only method that did not experience improvements in accuracy scores when employing encoded files. These factors lead us to believe that bzlib and zlib are probably more suitable for the zgli encoder module and agglomerative clustering than the lzma compression technique."

%%%%%%%%%%%%%%%%%%%%%%%%%%%%%%%%%%%%%%%%%%%%%%%%%%%%%%%%%%%%%%%%%%%%%%%%%%%%%%%%%%%%%%%%%%%%%%%%%%%%%%%%%%%%%%
* The findings summarized in lines 391-397 are not directly related to the results in tables 7, 8, & 9. The authors described the process but didn't show how this was applied to the data. If this is too large to be included in the article due to space restrictions, please consider adding them as supplementary material.

Our answer:

We thank the reviewer for pointing this out. We addressed this comment in the paragraph starting in line 444, where we talked about the patterns extracted from looking into the feature distribution of the clusters from our best models.

"Patterns from both clustering approaches were extracted by looking at the feature distribution of the clusters of the best models. These clusters showed that treatment response failed mainly when ASDAS values were close to low disease activity (2.1 to 2.7 threshold) and were successful when ASDAS values were high to very high disease activity values (3.1 upwards). Treatment response in the interval from 2.7 to 3.1 was less predictable. Still, gender showed to be correlated to this response, with Infliximab, Golimumab, and Etanercept treatments having a higher success rate for male patients and a higher failure rate for female patients. The Adalimumab treatment was the sole exception to this rule, having similar success rates for both male and female patients."

Round 2

Reviewer 3 Report

a missing reference or an incorrect symbol [?] in table 1 and the referenced text on lime 233. the same on lines 388, 397, and many more. please review and correct.

Author Response

We thank the reviewer for his/her sympathy toward our paper's publication. We also thank and apologize for the missing references. We correct the paper accordingly.